# DynGFN: Towards Bayesian Inference of Gene Regulatory Networks with GFlowNets

**Lazar Atanackovic**[1,2*]    **Alexander Tong**[3,4*]    **Bo Wang**[1,2,5]    **Leo J. Lee**[1,2]

**Yoshua Bengio**[3,4,6]    **Jason Hartford**[3,4,7]

[1]University of Toronto, [2]Vector Institute
[3]Mila - Quebec AI Institute, [4]Université de Montréal
[5]University Health Network, [6]CIFAR Fellow, [7]Valence Labs

## Abstract

One of the grand challenges of cell biology is inferring the gene regulatory network (GRN) which describes interactions between genes and their products that control gene expression and cellular function. We can treat this as a causal discovery problem but with two non-standard challenges: **(1)** regulatory networks are inherently cyclic so we should not model a GRN as a directed acyclic graph (DAG), and **(2)** observations have significant measurement noise, so for typical sample sizes there will always be a large equivalence class of graphs that are likely given the data, and we want methods that capture this uncertainty. Existing methods either focus on challenge **(1)**, identifying *cyclic* structure from dynamics, or on challenge **(2)** learning complex Bayesian *posteriors* over DAGs, but not both. In this paper we leverage the fact that it is possible to estimate the "velocity" of gene expression with *RNA velocity* techniques to develop an approach that addresses both challenges. Because we have access to velocity information, we can treat the Bayesian structure learning problem as a problem of sparse identification of a dynamical system, capturing cyclic feedback loops through time. Since our objective is to model uncertainty over discrete structures, we leverage Generative Flow Networks (GFlowNets) to estimate the posterior distribution over the combinatorial space of possible sparse dependencies. Our results indicate that our method learns posteriors that better encapsulate the distributions of cyclic structures compared to counterpart state-of-the-art Bayesian structure learning approaches.

## 1   Introduction

Inferring gene regulatory networks (GRNs) is a long standing problem in cell biology [25, 44]. If we knew the GRN, it would dramatically simplify the design of biological experiments and the development of drugs because it would serve as a map of which genes to perturb to manipulate protein and gene expression. GRNs concisely represent the complex system of directed interactions between genes and their regulatory products that govern cellular function through control of RNA (gene) expression and protein concentration. We can treat GRN inference as a causal discovery problem by treating the regulatory structure between genes (variables) as causal dependencies (edges) that we infer / rule out by using gene expression data. Structure learning methods aim to automate this task by inferring a set of directed acyclic graphs (DAGs) that are consistent with the conditional independencies that we can measure among the variables [14, 41, 42]. While there may be multiple

---

*Equal Contribution
Correspondence to: (`l.atanackovic@mail.utoronto.ca`)

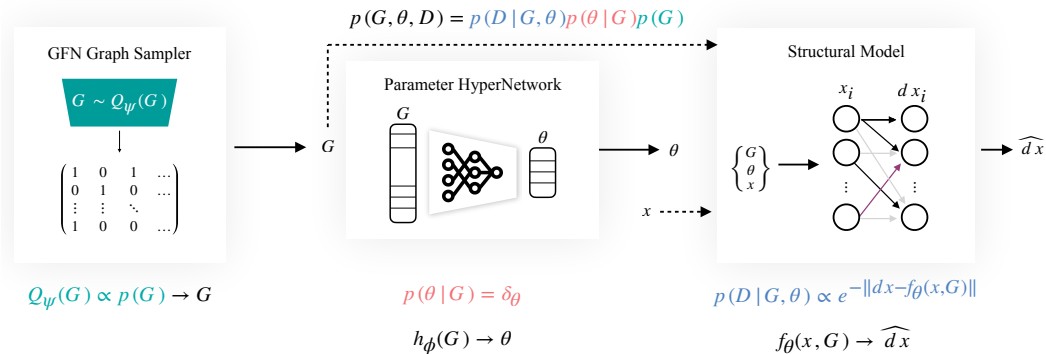

Figure 1: Architecture for Bayesian structure learning of dynamical systems. DynGFN consists of three main components: A GFlowNet modeling a posterior distribution over graphs $Q_\psi(G|\mathcal{D})$, a HyperNetwork modeling a posterior over parameters given a graph $Q_\phi(\theta|G, D)$, and the structural equation model scoring $G$ and $\theta$ according to how well they fit the data. Although the figure shows the case where $Q_\psi(G|\mathcal{D})$ is modelled with a GFlowNet, this can be any arbitrary graph sampler that can sample discrete structures $G \sim Q_\psi(G|\mathcal{D})$.

DAGs in this set—the "Markov equivalence class"—when we are able to perturb the variables with enough experimental interventions, it is possible to uniquely identify a causal graph [18].

However, structure learning for inferring GRNs comes with two non-standard challenges: **(1)** gene regulation contains inherent cyclic feedback mechanisms, hence we should not model a GRN as a DAG, and **(2)** observations are limited and have significant measurement noise, hence there exists a large equivalence class of graphs that are likely given datasets with typical sample sizes. Existing methods either focus on **(1)** – identifying graphs with *cyclic* structure by leveraging dynamics [15, 13] or assuming the system is in equilibrium [36], or **(2)** – learning complex Bayesian *posteriors* over explanatory DAGs [12], but not both. In this work, we address both challenges concurrently in a fully differentiable end-to-end pipeline (see Figure 1).

To accomplish this, we treat structure learning as a problem of sparse identification of a dynamical system. From a dynamical systems perspective, one can model both causal structure between variables as well as their time-dependent system response with the drift function [37, 43]. We leverage the fact that we can estimate the rate of change of a gene's expression (velocity) with *RNA velocity* methods [9]. This data takes the form of dynamic tuple pairs $(x, dx)$, which we can use to pose the dynamical system learning problem as a regression task (see Figure 1). This significantly simplifies the learning objective as we can model system dynamics while also learning structure without the need for numerically intensive differential equation solvers. We view this as a step towards Bayesian structure learning from continuous dynamics – we term this *Bayesian dynamic structure learning*.

Our approach estimates the posterior over the sparse dependencies and parameters of the dynamical system. This is important in scientific applications because it is usually prohibitively expensive to acquire a enough data to uniquely identify the true graph underlying a data generating process. Capturing the complex distribution over candidate structure is critical for downstream scientific applications and is an essential step in active causal discovery [39, 52, 19]. This is especially important in settings where experiments are expensive, e.g. conducting genetic perturbations for inference of GRNs. *Bayesian structure learning* is a class of methods that try to model this distribution over structure from observed data. These methods model posteriors over admissible structures $P(G|D)$ that explain the observations [32, 11, 4, 12, 31], but focus on modelling distributions over DAGs.

Our approach leverages Generative Flow Networks (GFlowNets) to model complex distributions over *cyclic* structures. GFlowNets [7, 8] parameterize the distribution over any discrete object (e.g. graphs) through a sequential policy, and as a result avoid needing to make restrictive parametric assumptions on the distribution. This makes them a useful tool in structure learning, particularly in cases where $P(G|D)$ is discrete and complex [12]. In this work, we use GFlowNets to learn posteriors over the sparse structure in a dynamical system, and separately learn the posteriors over the parameters of the drift function via a HyperNetwork [16] that conditions on inferred structures. Our main contributions are summarized as follows:

- We develop a novel framework for Bayesian structure learning under the lens of dynamical system identification for modelling complex posteriors over cyclic graphs. We consider flexible parameterizations for the structural model such that we can capture both linear and non-linear dynamic relationships.

- We design a novel GFlowNet architecture, Dynamic GFlowNet (DynGFN), tailored for modelling posteriors over cyclic structures. We propose a *per-node* factorization within DynGFN that enables efficient search over the discrete space of cyclic graphs.

- We empirically evaluated DynGFN on synthetic dynamic data designed to induce highly multi-modal posteriors over graphs.

- We showcase the use of DynGFN on a real biological system using single-cell RNA-velocity data for learning posteriors of GRNs.

## 2 Related Work

There are many works on the problem of identifying causal structure $G$ from either observational [e.g. 49, 54, 35] or interventional [e.g. 26, 29, 38] data, but the majority of existing methods return only the most likely DAG under the observed data. By returning only the most likely graph, these methods are often overconfident in their predictions. Bayesian approaches attempt to explicitly model a posterior distribution over DAGs given the data and model specification.

**Bayesian Structure Learning:** Recently, there has been significant interest in fully differentiable Bayesian methods for structure learning in the static case. DiBS [32], BCD-Nets [11], VCN [4], and DAG-GFlowNet [12] all attempt to learn a distribution over structural models from a fully observed system. The key difference is in how these methods parameterize the graph. DiBS is a particle variational inference method that uses two matrices $U$ and $V$ where $G = \texttt{sigmoid}(U^T V)$ where the sigmoid is applied elementwise which is similar to graph autoencoders. BCD-Nets and DP-DAG use the Gumbel-Sinkhorn distribution to parameterize a permutation and direct parameterization of a lower triangular matrix. VCN uses an autoregressive LSTM to generate the graph as this gets rid of the standard uni-modal constraint of Gaussian distributed parameters. DAG-GFN has shown success for modelling $P(G|\mathcal{D})$ [12]. However, it remains restrictive to assume the underlying structure of the observed system is a DAG as natural dynamical systems typically contain regulating feedback mechanisms. This can be particularly challenging for GFlowNets since including cycles in the underlying structure exponentially increases the discrete search space. We show that under certain assumptions we can in part alleviate this shortcoming for learning Bayesian posteriors over cyclic structures for dynamical systems. In small graphs, these methods can model the uncertainty over possible models (including over Markov equivalence classes).

**Dynamic and Cyclic Structure Learning:** There has been comparatively little work towards Bayesian structure learning from dynamics. Recent works in this direction based on NeuralODEs [10] propose a single explanatory structure [50, 6, 1, 2]. CD-NOD leverages heterogeneous non-stationary data for causal discovery when the underlying generative process changes over time [53, 21]. A similar approach uses non-stationary time-series data for causal discovery and forecasting [20]. DYNOTEARS is a score-based approach that uses time-series to learn structure [40]. However, these methods do not attempt to explicitly model a distribution over the explanatory structure. Other methods aim to learn cyclic dependencies in the underlying graph [24, 36, 28, 3]. For instance, [24] propose an iterative method that leverages interventional data to learn directed cyclic graphs. It is suggested that CD-NOD is also extendable to learn cyclic structure [21]. But these methods do not model a posterior over structure. In general, there remains a gap for the problem of Bayesian structure learning over cyclic graphs.

We include further discussion on related work for GRN inference from single-cell transcriptomic data and cell dynamics in Appendix C.1.

# 3 Preliminaries

## 3.1 Bayesian Dynamic Structure Learning

**Problem Setup:** We consider a finite dataset, $\mathcal{D}$, of dynamic pairs $(x, dx) \in \mathbb{R}^d \times \mathbb{R}^d$ where $x$ respresents the state of the system sampled from an underlying time-invariant stochastic dynamical system governed by a latent drift $\frac{dx}{dt} = f(x, \epsilon)$ where $\epsilon$ is a noise term that parameterizes the SDE; $x$ and $\epsilon$ are mutually independent. The latent drift has some fixed sparsity pattern i.e. $\frac{\partial f_i}{\partial x_j} \neq 0$ for a small set of variables, which can be parameterized by a graph $G$ such that $g_{ij} = \mathbf{1}[\frac{\partial f_i}{\partial x_j} \neq 0]$, where $g_{ij} \in G, i = 1, \ldots, d, j = 1, \ldots, d$. The variables $x_j$ for which $\frac{\partial f_i}{\partial x_j} \neq 0$ can be interpreted as the causal parents of $x_i$, denoted $\mathrm{Pa}(x_i)$. This lets us define an equivalent dynamic structural model [37, 43] of the form,

$$\frac{dx_i(t)}{dt} = f_i(\mathrm{Pa}(x_i), \epsilon_i), \tag{1}$$

for $i = 1, \ldots, d$. For the graph $G$ to be identifiable, we assume that all relevant variables are observed, such that *causal sufficiency* is satisfied.

Our goal is to model our posterior over explanatory graphs $Q(G|\mathcal{D})$ given the data. We aim to jointly learn distribution over parameters $\theta$ that parameterize the latent drift $f(x)$; these parameters will typically depend on the sparsity pattern such that $p(\theta|G) \neq p(\theta)$. We can factorize this generative model as follows,

$$p(G, \theta, \mathcal{D}) = p(\mathcal{D}|G, \theta)p(\theta|G)p(G) \tag{2}$$

This factorization forms the basis of our inference procedure. We learn a parameterized function $f_\theta(x) : \mathbb{R}^d \to \mathbb{R}^d$ that approximates the structural model defined in (1). To model this joint distribution, we need a way of representing $P(G)$, a distribution over the combinatorial space of possible sparsity patterns, and $P(\theta|G)$, the posterior over the parameters of $f_\theta$. We use GFlowNets [7] to represent $P(G)$, and a HyperNetwork to parameterize $P(\theta|G)$.

## 3.2 Generative Flow Networks

GFlowNets are an approach for learning generative models over spaces of discrete objects [7, 8]. GFlowNets learn a stochastic policy $P_F(\tau)$ to sequentially sample an object $\mathbf{x}$ from a discrete space $\mathcal{X}$. Here $\tau = (s_0, s_1, \ldots, s_n)$ represents a full Markovian trajectory over plausible discrete states, where $s_n$ is the terminating state (i.e. end of a trajectory) [34]. The GFlowNet is trained such that at convergence, sequential samples from the stochastic policy over a trajectory, $\mathbf{x} \sim P_F(\tau)$, i.e. $\mathbf{x} = s_n$, are equal in distribution to samples from the normalized reward distribution $P(\mathbf{x}) = \frac{R(\mathbf{x})}{\sum_{\mathbf{x}' \in \mathcal{X}} R(\mathbf{x}')}$. The GFlowNet policies are typically trained by optimizing either the *Trajectory Balance* (TB) loss [34], *Subtrajectory Balance* (Sub-TB) loss [33], or the *Detailed Balance* (DB) loss [12]. In this work, we exploit the DB loss to learn a stochastic policy for directed graph structure.

**Detailed Balance Loss:** The DB loss [12] leverages the fact that the reward function can be evaluated for any partially constructed graph (i.e. any prefix of $\tau$), and hence we get intermediate reward signals for training the GFlowNet policy. The DB loss is defined as:

$$\mathcal{L}_{\mathrm{DB}}(s_i, s_{i-1}) = \left( \log \frac{R(s_i)P_B(s_{i-1}|s_i; \psi)P_F(s_n|s_{i-1}; \psi)}{R(s_{i-1})P_F(s_i|s_{i-1}; \psi)P_F(s_n|s_i; \psi)} \right)^2, \tag{3}$$

where $P_F(s_i|s_{i-1}; \psi)$ and $P_B(s_{i-1}|s_i; \psi)$ represent the forward transition probability and backward transition probability, and a trainable normalizing constant, respectively. Under this formulation, during GFlowNet training the reward is evaluated at every state. For this reason, the DB formulation is in general advantageous for the structure learning problem where any sampled graph can be viewed as a complete state, hence more robustly inform gradients when training the stochastic policy than counterpart losses. Previous work has shown GFlowNets are useful in settings with multi-modal posteriors. This is of particular interest to us where many admissible structures can explain the observed data equally well. We model $Q_\psi(G)$ using $P_F(s_i|s_{i-1}; \psi)$ and learn the parameters $\psi$.

---
**Algorithm 1** Batch update training of DynGFN
---
1: **Input:** Data batch $(x_b, dx_b)$, initial NN weights $\psi, \phi$, $L^0$ sparsity prior $\lambda_0$, and learning rate $\epsilon$.
2: $s_0 \leftarrow \mathbf{0}_{B \times d \times d}$             ▷ *Training is paralleled over $B$ graph trajectories*
3: $a \sim P_F(s_1|s_0; \psi)$,                  ▷ *Sample initial actions vector*
4: **while** $a$ not $\emptyset$ **do**
5:      Compute $P_F(s_i|s_{i-1}; \psi), P_B(s_{i-1}|s_i; \psi)$
6:      $\theta \leftarrow h_\phi(s_i)$
7:      $\widehat{dx_b} \leftarrow f_\theta(x, s_i)$
8:      $R_i(s_i) \leftarrow e^{-\|dx_b - \widehat{dx_b}\|_2^2 - \lambda_0 \|s_i\|_0}$
9:      $\psi \leftarrow \psi - \epsilon \nabla_\psi \mathcal{L}_{DB}(s_i, s_{i-1})$       ▷ $\mathcal{L}_{DB}(s_i, s_{i-1})$ *computed as in Equation 3*
10:      $a \sim P_F(s_i|s_{i-1}; \psi), s_i \rightarrow s_{i+1}$        ▷ *Take action step to go to next state*
11: $\phi \leftarrow \phi + \epsilon \nabla_\phi \log R$
     **return** Updated GFN weights $\psi$ and updated HyperNetwork weights $\phi$.
---

## 4 DynGFN for Bayesian Dynamic Structure Learning

We present a general framework for Bayesian dynamic structure learning and propose a GFlowNet architecture, *DynGFN*, tailored for modelling a posterior over discrete cyclic graphical structures. We summarize our framework in Figure 1 and Algorithm 1. DynGFN consists of 3 key modules:

1. A graph sampler that samples graphical structures that encode the structural dependencies among the observed variables. This is parameterized with a GFlowNet that iteratively adds edges to a graph.

2. A model that approximates the structural equations defined in (1) to model the functional relationships between the observed variables, indexed by parameters $\theta$. This is a class of functions that respect the conditional independencies implied by the graph sampled in step 1. We enforce this by masking inputs according to the graph.

3. Because the functional relationships between variables may be different depending on which graph is sampled, we use a HyperNetwork architecture that outputs the parameters $\theta$ of the structural equations as a function of the graph. We also show that under linear assumptions of the structural modules, we can solve for optimal $\theta$ analytically (i.e. we do not need the HyperNetwork).

For training, we assume $L^0$ sparsity of graphs $G$ to constrain the large discrete search space over possible structures. We use a reward $R$ for a graph $G$ and $L^0$ penalty of the form: $R(G) = e^{-\|dx - \widehat{dx}\|_2^2 + \lambda_0 \|G\|_0}$. We motivate this set-up so we can estimate $\widehat{dx}$ close to $dx$ in an end-to-end learning pipeline. Since estimates for $\widehat{dx}$ are dependent on $G$ and $\theta$, this reward informs gradients to learn a policy that can approximate $Q(G)$ given dynamic data.

The main advantage of DynGFN comes when modelling complex posteriors with many modes. Prior work has shown GFlowNets are able to efficiently model distributions where we can share information between different modes [34]. The challenge we tackle is how to do this with a changing objective function, as the GFlowNet objective is a function of the current parameter HyperNetwork and the structural equations. We use multilayer perceptrons (MLPs) to parameterize the stochastic GFlowNet policy, HyperNetwork architecture, and the dynamic structural model[1].

### 4.1 Graph Sampler

DynGFN models a posterior distribution over graphs $Q(G|\mathcal{D})$ given a finite set of observations. To learn $Q(G|\mathcal{D})$, DynGFN needs to explore over a large discrete state space. Since we aim to learn a bipartite graph between $x$ and $dx$, DynGFN needs to search over $2^{d^2}$ possible structures, where $d$ denotes the dimensionality of the system and $2^{d^2}$ the number of possible edges in $G$. For even moderate $d$, this discrete space is very large (e.g. for $d = 20$ we have $2^{400}$ possible graphs).

---
[1]When we assume linear dynamic structural relationships, we can solve for the parameters analytically, thus do not need MLPs for the HyperNetwork and dynamic structural model. This is further discussed in section 4.2

However, under the assumption of causal sufficiency, we can significantly reduce this search space, by taking advantage of the fact that $Q(G|\mathcal{D})$ factorizes as follows,

$$Q(G|D) = \prod_{i \in [1,...,d]} Q_i(G[\cdot, i]|D) \qquad (4)$$

By using this model, we reduce the search space from $2^{d^2} \to d2^d$. For $d = 20$ this reduces the search space from $2^{400}$ to $\approx 2^{24.3}$. While still intractable to search over, it is still a vast improvement over the unfactorized case. We call this model a *per-node* posterior, and we use a per-node GFlowNet going forward. We discuss details regarding encouraging forward policy exploration during training in Appendix B.6.

## 4.2 HyperNetwork and Structural Model

We aim to jointly learn the structural encoding $G$ and parameters $\theta$ that together model the structural relationships $dx = f_\theta(x, G)$ of the dynamical system variables. To accomplish this, we propose learning an individual set of parameters $\theta$ for each graph $G$, independent of the input data $x$. This approach encapsulates $P(\theta|G)$ in (2). We use a HyperNetwork architecture that takes $G$ as input and outputs the structural equation model parameters $\theta$, i.e. $\theta = h_\phi(G)$ hence $P(\theta|G) = \delta(\theta|G)$ – allowing us to learn a separate $\theta$ for each $G$. This HyperNetwork model does not capture uncertainty in the parameters, however the formulation may be extended to the Bayesian setting by placing a prior on the HyperNetwork parameters $\phi$. Although $h_\phi$ allows for expressive parameterizations for $\theta$, it may not be easy to learn[2]. HyperNetworks have shown success in learning parameters for more complex models (e.g. LSTMs and CNNs) [16], hence motivates their fit for our application.

**Linear Assumption on Dynamic Structural Model:** In some cases it may suffice to assume a linear differential form $\frac{dx}{dt} = \boldsymbol{A}x$ to approximate dynamics. In this setting, given a sampled graph $G \sim Q(G)$ and $n$ i.i.d. observations of $(x, dx)$ we can solve for $\theta = \boldsymbol{A}$ analytically. To induce dependence on the graph structure, we use the sampled $G$ as a mask on $x$ and construct $\tilde{x}_i = G_i^T \odot x$. Then we can solve for $\theta$ on a per-node basis as

$$\theta_i = (\tilde{x}_i^T \tilde{x}_i + \lambda I)^{-1} \tilde{x}_i^T dx_i, \qquad (5)$$

where $i = 1, \ldots d$, $\lambda > 0$ is the precision of an independent Gaussian prior over the parameters, and $I$ is the identity matrix. We use $\lambda = 0.01$ throughout this work.

## 5 A Useful Model of Indeterminacy

In order to evaluate the ability of DynGFN to model complex posteriors over graphs, we need a structure learning problem with a large equivalence class of admissible graphs. We present a simple way to augment a set of identifiable dynamics under some model to create a combinatorial number of equally likely dynamics under the same model. More specifically, this creates a ground truth posterior $Q^*(G|D) \propto \sum T(G^*)$ where $T(\cdot) : \mathcal{G} \to \mathcal{G}$ is an analytically computable transformation over graphs and $G^*$ is the identified graph under the original dynamics. We use this system to test how well we can learn a posterior over structures that matches what we see in single-cell data.

Specifically, given a dataset of $(x, dx) \in \mathbb{R}^d \times \mathbb{R}^d$ pairs, we create a new dataset with $d+1$ variables where the 'new' variable $v'$ is perfectly correlated with an existing variable $v$. In causal terms, this new variable inherits the same parents as $v$, that is $\text{Pa}(v') := \text{Pa}(v)$ and the same structural equations as $v$, that is $dv' = dv$. This is depicted in Figure 2. This creates a number of new possible explanatory graphs, which we generalize with the following proposition.

**Proposition 1.** *Given any $d$ dimensional ODE system with $\mathcal{G}^*$ identifiable under $f \in \mathcal{F}$, the $D = d + a$ dimensional system $\frac{dx}{dt} = \boldsymbol{A}x$, denote the vector of multiplicities $m \in \mathbb{N}^d$ with $m_i$ as the number of repetitions of each variable. Then this construction creates an admissible family of graphs $\mathcal{G}'$ where $|\mathcal{G}'| = \prod_{i \in d}(2^{m_i} - 1)^{\mathcal{G}_i \mathbf{1}}$. Furthermore, under an $L^0$ penalty on $G$, this reduces to $\prod_i (m_i)^{\mathcal{G}_i \mathbf{1}}$.*

See Appendix A for full proof. The intuition behind this proposition can be seen from the case of adding a single copied variable. This corresponds to $\boldsymbol{A} = [\delta_v \boldsymbol{I}_d]$ where $\delta_v$ is a vector with a 1 on

---

[2]We discuss training dynamics when using $h_\phi$ in Appendix B.7.

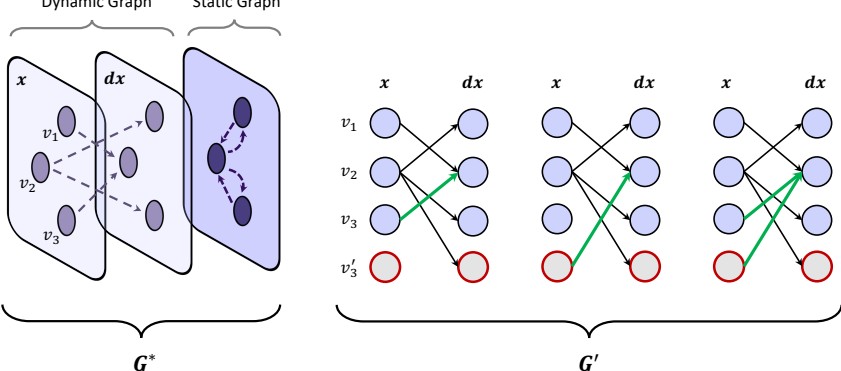

Figure 2: Visualization of modeling cyclic dependencies over time (left). To create a family of admissible graphs (right), we add a new variable $v_3'$ which has the same values as $v_3$ and creates three possible explanations for the data (green arrows). If we apply a sparsity penalty, then we can eliminate the last possibility (which has two additional edges) for only two possible graphs. $G^*$ denotes the ground truth graph while $G'$ denotes the admissible family of graphs induced by $v_3'$.

node $v$ and zeros elsewhere, and $\boldsymbol{I}_d$ is the $d$-dimensional identity matrix. Let $v$ have $c$ children, such that $v \in Pa(c)$ in the identifiable system, then any of those $c$ child nodes could depend either on $v$ or on the new node $v'$ or both. This creates $3^c$ possible explanatory graphs. If we restrict ourselves to the set of graphs with minimal $L^0$ norm, then we eliminate the possibility of a child node depending on both $v$ and $v'$, this gives $2^c$ possible graphs, choosing either $v$ or $v'$ as a parent.

## 6    Experimental Results

In this section we evaluate the performance of DynGFN against counterpart Bayesian structure learning methods (see Appendix B.2 for details). Since our primary objective is to learn Bayesian posteriors over discrete structure $G$, we compare to Bayesian methods that can also accomplish this task, i.e. versions of BCD-Nets [11] and DiBS [32]. We show in certain cases, DynGFN is able to better capture the true posterior when there are a large number of modes. We evaluate methods according to four metrics: Bayes-SHD, area under the receiver operator characteristic curve (AUC), Kullback–Leibler (KL) divergence between learned posteriors $Q(G)$ and the distribution over true graphs $P(G^*)$, and the negative log-likelihood (NLL) $P(D|G, \theta)$ (in our setting this reduces to the mean squared error between $\widehat{dx}$ and $dx$, given $\theta$ and sampled $G's$). Since the analytic linear solver requires data at run-time to compute optimal parameters for the structural model, we include the NLL metric only for models using the HyperNetwork solver. Bayes-SHD measures the average distance to the closest structure in the admissible set of graphs according to the structural hamming distance, which in this case is simply the hamming distance of the adjacency matrix representation to the closest admissible graph. We assume $P(G^*)$ is uniform over $G^*$ and include further details about evaluating the quality of learned posteriors in Appendix B.8.

### 6.1    A Toy Example with Synthetic Data

To validate the ability of DynGFN to learn cyclic dependencies we consider identifiable acyclic and cyclic 3 variable toy systems and provide a comparison with a DAG structure learning method (NOTEARS [54]). We show results for this toy example in Figure 3. NOTEARS does not model cyclic dependencies and therefore struggles to yield accurate predictions of $\boldsymbol{A}$ in the cyclic setting. We can also verify this result by considering a conditional independence test over cyclic dependencies. It is easy to see that the conditional independence test fails in the cyclic setting: in the acyclic case, we can identify the v-structure by observing that $x_1 \perp x_3$ and $x_1 \not\perp x_3|x_2$, which implies that $x_2$ is a collider (i.e. $x_1$ and $x_3$ are marginally independent and conditionally dependent); while in the cyclic example, we introduce time dependencies such that there are cycles in the summary graph that render these variables marginally dependent. We show that DynGFN is able to identify the true cyclic dependencies in this toy example. We note that in this toy example DynGFN exhibits a degree

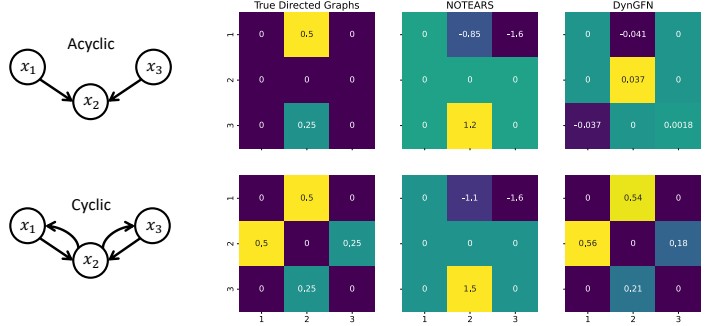

Figure 3: Toy example for learning DAG structure (top) and cyclic graph structure (bottom) from observational data. In this example we compare NOTEARS and DynGFN for learning acyclic and cyclic dependencies from observational data. Data is generated from a linear system $dx = \boldsymbol{A}x$ with a corresponding acyclic and cyclic $\boldsymbol{A}$ as defined in the figure. We use $500$ observational samples for each system. NOTEARS is implemented using the CausalNex [5] library.

Table 1: Bayesian dynamic structure learning of linear and non-linear systems with $d = 20$ variables. The graphs representing the structural dynamic relationships of the linear and non-linear systems have $50$ edges out of possible $400$. The ground truth discrete distribution $P(G^*)$ contains $1024$ admissible graphs for each respective system. The $\ell$ and $h$ pre-fix denote usage of the analytic linear solver and HyperNetwork solver for structural model parameters, respectively. Results are reported on held out test data over 5 model seeds.

| | **Linear System** | | | |
| **Model** | **Bayes-SHD** $\downarrow$ | **AUC** $\uparrow$ | **KL** $\downarrow$ | **NLL** $\downarrow$ |
| --- | --- | --- | --- | --- |
| $\ell$-DynBCD | $32.0 \pm 0.27$ | $0.71 \pm 0.0$ | $1707.45 \pm 9.66$ | — |
| $\ell$-DynDiBS | $29.2 \pm 0.78$ | $0.71 \pm 0.0$ | $6622.43 \pm 171.67$ | — |
| $\ell$-DynGFN | $\mathbf{22.8 \pm 1.4}$ | $\mathbf{0.75 \pm 0.01}$ | $\mathbf{1091.60 \pm 35.72}$ | — |
| $h$-DynBCD | $\mathbf{5.5 \pm 1.1}$ | $0.89 \pm 0.04$ | $701.19 \pm 46.99$ | $(9.83 \pm 0.59)\mathrm{E}-5$ |
| $h$-DynDiBS | $28.5 \pm 4.2$ | $0.51 \pm 0.07$ | $7934.90 \pm 381.80$ | $(\mathbf{8.17 \pm 1.30})\mathrm{E}-6$ |
| $h$-DynGFN | $\mathbf{6.7 \pm 0.0}$ | $\mathbf{0.94 \pm 0.0}$ | $\mathbf{350.92 \pm 30.15}$ | $(8.35 \pm 0.02)\mathrm{E}-3$ |

| | **Non-linear System** | | | |
| **Model** | **Bayes-SHD** $\downarrow$ | **AUC** $\uparrow$ | **KL** $\downarrow$ | **NLL** $\downarrow$ |
| --- | --- | --- | --- | --- |
| $\ell$-DynBCD | $77.5 \pm 8.3$ | $0.42 \pm 0.03$ | $3814.86 \pm 354.56$ | — |
| $\ell$-DynDiBS | $75.7 \pm 7.7$ | $\mathbf{0.59 \pm 0.01}$ | $5893.65 \pm 59.66$ | — |
| $\ell$-DynGFN | $\mathbf{45.7 \pm 0.6}$ | $0.55 \pm 0.0$ | $\mathbf{226.25 \pm 6.58}$ | — |
| $h$-DynBCD | $192.9 \pm 0.7$ | $0.50 \pm 0.0$ | $9108.69 \pm 51.34$ | $(3.83 \pm 0.32)\mathrm{E}-4$ |
| $h$-DynDiBS | $48.1 \pm 9.0$ | $0.53 \pm 0.10$ | $8716.64 \pm 265.29$ | $(\mathbf{4.06 \pm 0.10})\mathrm{E}-6$ |
| $h$-DynGFN | $\mathbf{32.6 \pm 0.9}$ | $\mathbf{0.67 \pm 0.01}$ | $\mathbf{193.28 \pm 8.53}$ | $(1.47 \pm 0.11)\mathrm{E}-3$ |

of convergence sensitivity on a per-run basis. In the following sections we provide comprehensive results over 5 random seeds and consider larger systems.

## 6.2   Experiments with Synthetic Data

We generated synthetic data from two systems using our indeterminacy model presented in section 5: (1) a linear dynamical system $dx = \boldsymbol{A}x$, and (2) a non-linear dynamical system $dx = \texttt{sigmoid}(\boldsymbol{A}x)$. We consider $\ell$-DynGFN and $h$-DynGFN, i.e. DynGFN with the linear analytic parameter solver as shown in (5), and DynGFN with the HyperNetwork parameter solver $h_\phi$. Likewise, we compare $\ell$-DynGFN and $h$-DynGFN to counterpart Bayesian baselines which we call $\ell$-DynBCD, $\ell$-DynDiBS, $h$-DynBCD, and $h$-DynDiBS. To constrain the discrete search procedure, we assume a sparse prior on the structure (i.e. the graphs $G$), using the $L^0$ prior. Due to challenging iterative optimization dynamics present when using $\theta = h_\phi(G)$ for DynGFN, to train initialize the forward policy $P_F(s_i|s_{i-1}; \psi)$ using the $\psi$ learned in $\ell$-DynGFN to provide a more admissible starting point for learning $h_\phi$ (we

discuss further details in Appendix B.7). We do not need to do this for $h$-DynBCD and $h$-DynDiBS as we are able to train both models end-to-end without iterative optimization. In Table 1 we show results of our synthetic experiments for learning posteriors over multi-modal distributions of cyclic graphs. We observe the DynGFN is most competitive on both synthetic systems for modelling the true posterior over structure. Details about DynGFN, baselines, and accompanying hyper-parameters can be found in Appendix B.

## 6.3 Ablations Over Sparsity and Linearity of Dynamic Systems

We conduct two ablations: (1) ablation over sparsity of the dynamic system structure, and (2) ablation over $\Delta t$, the time difference between data points of dynamic simulation. For a sparsity level of 0.9, the ground truth graphs have 50 edges out of $d^2$ possible edges. In these experiments, $P(G^*)$ for $d = 20$ and sparsity 0.9 has 1024 modes. We conduct the ablations over 5 random seeds for each set of experiments.

**Sparsity:** DynGFN uses the $L^0$ prior on $G$ throughout training. Under this setting, system sparsity carries significant weight on the ability to learn posteriors over the structured dynamics of a system. We show this trend in Table 2. We note that computing the KL-divergence for DynGFN, specifically computing the probability of generating a true $G$, becomes computationally intractable as $G$ is less sparse[3]. For systems of 0.9 and 0.95 sparsity, we observe a decreasing trend in KL and Bayes-SHD, and an increasing trend in AUC. This result is expected as DynGFN can better traverse sparse graphs as the combinatorial space over possible trajectories is smaller relative to denser systems.

Table 2: Ablation for $\ell$-DynGFN on $d = 20$ systems with varying levels of sparsity and fixed $\Delta t = 0.05$.

| Sparsity | Bayes-SHD ↓ | AUC ↑ | KL ↓ |
|---|---|---|---|
| 0.95 | $16.4 \pm 1.71$ | $0.79 \pm 0.0$ | $889.57 \pm 31.24$ |
| 0.90 | $22.8 \pm 1.41$ | $0.75 \pm 0.01$ | $1091.60 \pm 35.72$ |
| 0.85 | $32.8 \pm 0.72$ | $0.71 \pm 0.0$ | — |
| 0.80 | $39.2 \pm 0.69$ | $0.71 \pm 0.0$ | — |
| 0.75 | $60.2 \pm 1.17$ | $0.66 \pm 0.01$ | — |

Table 3: Ablation for $\ell$-DynGFN on $d = 20$ non-linear systems with varying $\Delta t$ and fixed sparsity at 0.9.

| $\Delta t$ | Bayes-SHD ↓ | AUC ↑ | KL ↓ |
|---|---|---|---|
| 0.001 | $38.7 \pm 0.80$ | $0.61 \pm 0.0$ | $202.41 \pm 9.95$ |
| 0.005 | $39.0 \pm 0.81$ | $0.60 \pm 0.0$ | $206.83 \pm 11.55$ |
| 0.01 | $40.6 \pm 1.13$ | $0.59 \pm 0.0$ | $202.71 \pm 7.74$ |
| 0.05 | $45.7 \pm 0.62$ | $0.55 \pm 0.0$ | $226.25 \pm 6.58$ |
| 0.1 | $51.8 \pm 0.18$ | $0.50 \pm 0.0$ | $264.86 \pm 2.17$ |

**Linearity:** Training DynGFN via the linear solver for the structural model parameters is an easier objective due to simplified training dynamics. Because of this, we explore the performance of $\ell$-DynGFN assuming $f_\theta$ for modelling equation (1) to be linear in the non-linear system. We do this by conducting an ablation over $\Delta t$ and find that the performance of $\ell$-DynGFN on the non-linear system improves as $\Delta t \to 0$. We show a portion of this trend in Table 3.

## 6.4 Experiments on Single-Cell RNA-velocity Data

To show how DynGFN can be applied to single cell data we use a cell cycle dataset of human Fibroblasts [46]. As a motivating example we show the correlation structure of single-cell RNA-seq data from human Fibroblast cells [46] Figure 4. We show both the raw correlation and the correlation over cell cycle time, which is significantly higher. With such a pure cell population whose primary axis of variation is state in the cell cycle by aggregating over cell cycle time we expect observation noise to be averaged out, leading to a "truer" view of the correlation between latent variables. Further details for this experimental set-up are provided in Appendix C.1.3. Since there are many genes which are affected by the cell cycle phase, there are many correlated variables that are downstream of the true cell cycle regulators. This provides a natural way of using cell cycle data to evaluate a model's ability to capture the Bayesian posterior. In Table 4 we show results for learning posteriors over an undetermined GRN using RNA velocity data. We find that $\ell$-DynGFN and $h$-DynGFN yield low KL

---

[3]For example, since DynGFN constructs one object $G$ sequentially over a state space distribution, we must compute probabilities of all combinatorial state trajectories for constructing $G = (s_i, \ldots, s_n)$. The space of combinatorial state trajectories is $n!$ in nature, hence this computation is only possible for small graphs and/or sparse graphs.

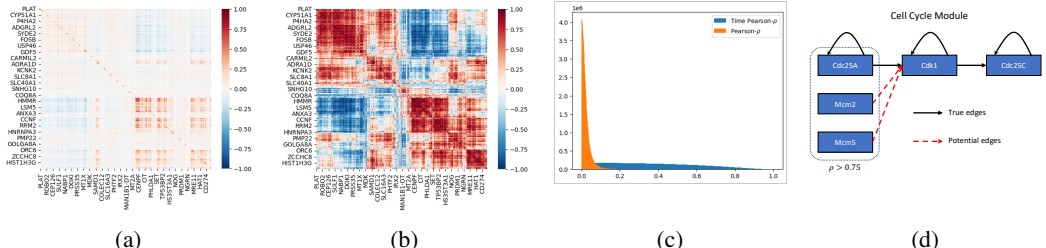

|  |  |  |  |
|---|---|---|---|
| (a) | (b) | (c) | (d) |

Figure 4: (a) Correlation structure in the raw single cell data over 5000 cells and 2000 genes selected by scVelo [8] pre-processing. (b) Correlation structure among genes over (inferred) cell cycle times. This stronger correlation structure is more reflective of the correlation in the underlying system. (c) Histogram of pairwise Pearson correlations between all genes passing pre-processing, comparing the absolute values of the elements in (a) and (b). (d) Shows the ground truth GRN extracted as a subset of the KEGG cell cycle pathway. Cdc25A is known to inhibit Cdk1 which is known to inhibit Cdc25C, while the Mcm complex is highly correlated with Cdc25A, they do not directly interact with Cdk1 [24].

Table 4: Bayesian dynamic structure learning 5-D cellular system using scRNA velocity data. The dynamics of this system are unknowns, however we identify 81 admissible graphs between variables (genes) that describe the data. We train models over 5 seeds. The graphs of this system contain of 7 true edges.

| | Cellular System - RNA Velocity | | | |
|---|---|---|---|---|
| **Model** | **Bayes-SHD** $\downarrow$ | **AUC** $\uparrow$ | **KL** $\downarrow$ | **NLL** $\downarrow$ |
| $\ell$-DynBCD | $\mathbf{2.6 \pm 0.1}$ | $0.56 \pm 0.01$ | $321.95 \pm 3.34$ | — |
| $\ell$-DynDiBS | $6.5 \pm 0.4$ | $0.47 \pm 0.01$ | $550.17 \pm 16.63$ | — |
| $\ell$-DynGFN | $\mathbf{3.3 \pm 0.4}$ | $\mathbf{0.59 \pm 0.03}$ | $\mathbf{44.98 \pm 18.60}$ | — |
| $h$-DynBCD | $10.1 \pm 0.8$ | $0.53 \pm 0.03$ | $587.41 \pm 24.00$ | $0.094 \pm 0.003$ |
| $h$-DynDiBS | $9.6 \pm 4.2$ | $0.51 \pm 0.13$ | $560.85 \pm 83.83$ | $\mathbf{0.084 \pm 0.0}$ |
| $h$-DynGFN | $\mathbf{5.1 \pm 1.2}$ | $\mathbf{0.58 \pm 0.05}$ | $\mathbf{39.82 \pm 28.05}$ | $0.109 \pm 0.001$ |

and moderate Bayes-SHD. While $\ell$-DynBCD performs well in terms of identify a small distribution of true $G$'s, it falls short in modelling the true posterior (this can be seen from low Bayes-SHD, high KL).

## 7 Conclusion

We presented DynGFN, a method for Bayesian dynamic structure learning. In low dimensions we found that DynGFN is able to better model the distribution over possible explanatory structures than counterpart Bayesian structure learning baseline methods. As a proof of concept, we presented an example of learning the distribution over likely explanatory graphs for linear and non-linear synthetic systems where complex uncertainty over explanatory structure is prevalent. We demonstrate the use of DynGFN for learning gene regulatory structure from single-cell transcriptomic data where there are many possible graphs, showing DynGFN can better model the uncertainty over possible explanations of this data rather than capturing a single explanation.

**Limitations and Future Work:** We have demonstrated a degree of efficacy when using DynGFN for Bayesian structure learning with dynamic observational data. A key limitation of DynGFN is scaling to larger systems. To effectively model $P(G, \theta, D)$, DynGFN needs to search over an environment state space of possible graphs. This state space grows exponentially with the number of possible edges, i.e. $2^{d^2}$ or $d2^d$ for per-node-GFN where $d$ is the number of variables in the system. Therefore, DynGFN is currently limited to smaller systems. Nevertheless, there are many applications where Bayesian structure learning, even over 5-20 dimensional examples that we explore here, could be extraordinarily useful. We include further discussion of scaling DynGFN in Appendix C.3 with some ideas on how to approach this challenge. We found that training DynGFN requires some selection of hyper-parameters and in particular parameters that shape the reward function. Selecting hyper-parameters for the baseline methods prove more difficult for this task.

# 8 Acknowledgments and Disclosure of Funding

This research was enabled in part by the computational researches provided by Mila (`mila.quebec`) and Compute Canada (`ccdb.computecanada.ca`). In addition, resources used in preparing this research were in part provided by the Province of Ontario and companies sponsoring the Vector Institute (`vectorinstitute.ai/partners/`). All authors are funded by their primary academic institution. We also acknowledged funding from the Natural Sciences and Engineering Research Council of Canada, Recursion Pharmaceuticals, CIFAR, Samsung, and IBM. We are grateful to Cristian Dragos Manta for catching some typos in the manuscript.

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

# Supplementary Material

## A   Proof of Proposition 1

Proposition 1 calculates the number of admissible structure graphs for a linear ODE system with correlated variables. We will first show the general case this is $\prod_{i \in d}(2^{m_i} - 1)^{\mathcal{G}_i \mathbf{1}}$, then analyze the case of an $L^0$ penalty on the edges of $G$, which reduces the size of the set of admissible graphs to $\prod_i (m_i)^{\mathcal{G}_i \mathbf{1}}$.

*Proof.* Consider an identifiable linear system $\frac{dx}{dt} = Ax$ where we directly observe $(x, \frac{dx}{dt})$ with $\mathcal{G}^*$ identifiable. Then the system with $m = \mathbf{1}^d$ has exactly one admissible graph by definition. For each node, we analyze its set of child nodes in $\mathcal{G}$, i.e. $c(u) = \{v \in \mathcal{V} \text{ s.t. } u \to v \in \mathcal{G}\}$. For an identifiable system, each child $v$ must have an incoming edge from its parent.

Next, we consider the process of adding a correlated variable, i.e. consider the situation of w.l.o.g. consider $m = (s, 1, 1, \ldots, 1)$ for some $s > 1$. Then for each child of $c_j(v_1)$, there are now $s$ possible parents. This has multiplied the number of possible graphs by $2^s - 1$. Since each element of $m$ is independent, this leads to the first statement, i.e. $|\mathcal{G}'| = \prod_{i \in d}(2^{m_i} - 1)^{\mathcal{G}_i \mathbf{1}}$.

Under an $L^0$ penalty, then we constrain the possible graphs to $s$ different graphs, where each child node picks exactly one of the $s$ possible parents. This leads to the second statement, $|\mathcal{G}'| = \prod_i (m_i)^{\mathcal{G}_i \mathbf{1}}$ $\qquad\qquad\square$

## B   Experimental Details

### B.1   Single Cell Dataset Preprocessing

We start with the processed data from [46]. We first filter it applying steps from the ScVelo tutorial. We then sub-select the genes of interest and use the "Ms" and "velocity" layers, which we normalize to mean zero standard deviation one for the states and scale the $dx$ with the same parameters.

### B.2   Baselines for Bayesian Dynamic Structure Learning

Existing Bayesian structure learning methods are typically constrained to learning DAGs. Temporal information about the the dynamic relationships amongst variables in a system can help alleviate this constraint. DiBS and BCD-Nets are two state-of-the-art Bayesian structure learning approaches for static systems. We apply versions of DiBS and BCD-Nets such that they are applicable in our Bayesian dynamic structure learning framework for learning cyclic graph structure from dynamic data. We use the approach taken in DiBS and parameterize the distribution over graphs as $P_{\alpha_t}(G|Z) = \prod_i \prod_j P_{\alpha_t}(G_{ij}|Z_{ij})$, where $Z = U^T V$, $U, V \in \mathbb{R}^{k \times d}$. Here $P_{\alpha_t}(G_{ij} = 1|Z_{ij}) = \sigma(\alpha_t Z_{ij})$, $\sigma(x) = 1/(1 + e^{-\alpha_t x})$, and $\alpha_t = \alpha c(t)$ ($t$ denotes the training iteration. We use $c(t) = \sqrt{t}$). As $t \to \infty$, $P_{\alpha_t}(G|Z) \to \delta(G|Z)$. In DiBS, Stein variational gradient decent (SVGD) [30] is used to iteratively transport particles $Z$ to learn the target distribution. Following from the above parameterization, we implement a version of BCD-Nets by treating $U \sim \mathcal{N}(\mu_{\mathbf{u}}, \sigma_{\mathbf{u}}^2), V \sim \mathcal{N}(\mu_{\mathbf{v}}, \sigma_{\mathbf{v}}^2)$, and learning $\mu_{\mathbf{u}}, \mu_{\mathbf{v}}, \sigma_{\mathbf{u}}$, and $\sigma_{\mathbf{v}}$. Since our framework uses dynamic data, we incorporate DiBS and BCD-Nets within the framework (labeled DynDiBS and DynBCD, respectively) to leverage dynamic information for Bayesian structure learning of cyclic graphs.

### B.3   Hyper-parameters for Baselines

For both DynBCD and DynDiBS we use $k = d$ across datasets. Since DynDiBS is an ensemble based method, we use 1024 samples for the linear and non-linear synthetic systems and 1000 samples for the cellular system (both training and evaluation). Since DynBCD is a variational approach and doesn't require parallelized model ensembles, we use a large quantity of samples for training and evaluation. In the case of the cellular system, since there is a significantly smaller quantity of admissible graphs, we use less samples for DynBCD. We use graph sparsity regularization denoted by $\lambda_0$ and a temperature parameter $T$ that scales the magnitude of the likelihood (e.g. $\frac{1}{T^2}\text{MSE}(dx, \widehat{dx})$).

In Table 5 and Table 6 we outline the hyper-parameters we found to yield the most competitive results. We use grid search to tune DynBCD and DynDiBS. All baselines are trained for 1000 epochs.

Table 5: Hyper-parameters for DynBCD. We define learning rate as $\epsilon$.

| | | **Linear System** | | | |
|---|---|---|---|---|---|
| **Model** | $\epsilon$ | $\lambda_0$ | $T$ | $\alpha$ | samples |
| $\ell$-DynBCD | 0.0001 | 0.001 | 0.01 | 0.1 | 5000 |
| $h$-DynBCD | 0.0001 | 0.0025 | 0.01 | 2 | 2000 |

| | | **Non-linear System** | | | |
|---|---|---|---|---|---|
| **Model** | $\epsilon$ | $\lambda_0$ | $T$ | $\alpha$ | samples |
| $\ell$-DynBCD | 0.00005 | 0.01 | 0.01 | 2 | 5000 |
| $h$-DynBCD | 0.0001 | 0.001 | 0.01 | 1 | 2000 |

| | | **Cellular System** | | | |
|---|---|---|---|---|---|
| **Model** | $\epsilon$ | $\lambda_0$ | $T$ | $\alpha$ | samples |
| $\ell$-DynBCD | 0.0001 | 0.001 | 0.05 | 0.05 | 1000 |
| $h$-DynBCD | 0.00001 | 0.0005 | 0.1 | 2 | 1000 |

Table 6: Hyper-parameters for DynDiBS. We define learning rate as $\epsilon$.

| | | **Linear System** | | | |
|---|---|---|---|---|---|
| **Model** | $\epsilon$ | $\lambda_0$ | $T$ | $\alpha$ | $\gamma$ |
| $\ell$-DynDiBS | 0.0025 | 500 | 0.01 | 0.0001 | 3000 |
| $h$-DynDiBS | 0.0001 | 3 | 0.01 | 0.0001 | 10000 |

| | | **Non-linear System** | | | |
|---|---|---|---|---|---|
| **Model** | $\epsilon$ | $\lambda_0$ | $T$ | $\alpha$ | $\gamma$ |
| $\ell$-DynDiBS | 0.001 | 10 | 0.01 | 0.0001 | 3000 |
| $h$-DynDiBS | 0.0001 | 0.1 | 0.01 | 0.0001 | 10000 |

| | | **Cellular System** | | | |
|---|---|---|---|---|---|
| **Model** | $\epsilon$ | $\lambda_0$ | $T$ | $\alpha$ | $\gamma$ |
| $\ell$-DynDiBS | 0.0025 | 1 | 0.05 | 0.0001 | 3000 |
| $h$-DynDiBS | 0.00001 | 0.1 | 0.01 | 0.01 | 3000 |

We note that when evaluation on validation and test data for Bayes-SHD and AUC metrics, we hard threshold $P_{\alpha_t}(G|Z)$. We find that through training this the final $\alpha_t$ is typically small enough in magnitude such that $P_{\alpha_t}(G|Z)$ does not yield a full threshold of $Z$. To this end, we select large $\alpha_t$ when computing the KL metric to mimic hard threshold behaviour as experienced during training. We use $\alpha_t = 1 \times 10^8$. In DynDiBS the parameter $\gamma$ helps control separation of particles $Z$ during training. In general, we found DynBCD and DynDiBS baselines are challenging to train and to find hyper-parameter settings with good performance. In part, we believe this is due to the numerous hyper-parameters required to tune as well as the general difficulty of the objective.

## B.4 Neural Network Architectures and Hyper-parameters

We parameterize $P_F(s_i|s_{i-1}; \psi)$ and $h_\phi$ with MLP architectures. $P_F(s_i|s_{i-1}; \psi)$ takes the current state as input and first computes common representations using a 3 layer MLP. Then a 2 layer MLP with a softmax output activation takes the representations as input and outputs a distribution over possible actions. The latter MLP is used to parameterize one head for each distribution $P_F(s_i|s_{i-1}; \psi)$. We use a hidden unit dimension of 128 and leaky rectified linear unit (Leaky ReLU) activation functions for the $P_F(s_i|s_{i-1}; \psi)$ MLP architecture. We use a uniform backward policy for $P_B(s_{i-1}|s_i; \psi)$. To parameterize $h_\phi$, we use a 3 layer MLP with hidden layer dimensions of

Table 7: Hyper-parameters for DynGFN. We define learning rate as $\epsilon$, $m_{train}$ as number of training samples, and $m_{eval}$ the number of evaluation samples.

**Linear System**

| Model | $\epsilon$ | $\lambda_0$ | $T$ | $m_{train}$ | $m_{eval}$ |
|---|---|---|---|---|---|
| $\ell$-DynGFN | 0.0001 | 100 | 0.01 | 1024 | 5000 |
| $h$-DynGFN | 0.00001 | 100 | 0.005 | 256 | 3000 |

**Non-linear System**

| Model | $\epsilon$ | $\lambda_0$ | $T$ | $m_{train}$ | $m_{eval}$ |
|---|---|---|---|---|---|
| $\ell$-DynGFN | 0.0001 | 150 | 0.01 | 1024 | 5000 |
| $h$-DynGFN | 0.00001 | 150 | 0.005 | 256 | 3000 |

**Cellular System**

| Model | $\epsilon$ | $\lambda_0$ | $T$ | $m_{train}$ | $m_{eval}$ |
|---|---|---|---|---|---|
| $\ell$-DynGFN | 0.00005 | 45 | 0.01 | 1024 | 1000 |
| $h$-DynGFN | 0.0001 | 10 | 0.1 | 1024 | 1000 |

$\{64, 64, 64\}$ and exponential linear unit activations (ELU). We consider two parametrizations for $f_\theta$: single linear parameters, i.e $dx = \theta x$, and a single hidden layer neural network $dx = f_\theta(x)$. We use these parametrizations to model linear and non-linear node-wise parent-child structural models, where $x \in \mathbb{R}^d$ are the node-wise input observations.

## B.5  Hyper-parameters for DynGFN

DynGFN requires setting a variety of hyper-parameters that lead to different trade offs in model performance. In particular, $\lambda_0$ (sparsity encouragement for identified graphs), a temperature parameter $T$ that scales the magnitude of the reward likelihood (e.g. $\frac{1}{T^2}\text{MSE}(dx, \widehat{dx})$), learning rate $\epsilon$, softmax tempering $c$ (we always use a cosine schedule for c, with a discrete period of 5 epochs), and number of training epochs. In our experiments we select hyper-parameter values that we find lead to competitive performance (this pertains to $\ell$-DynGFN and $h$-DynGFN models) by observing performance over a few hyper-parameter values. We outline the selected hyper-parameters for each respective model in Table 7. For GFlowNet sampling, due to computational limits, we use less training samples $m_{train}$ than evaluation samples $m_{eval}$ for DynGFN. We train DynGFN for 1000 epochs.

## B.6  GFlowNet Exploration vs. Exploitation

The general procedure for training GFlowNets is inspired from reinforcement learning where the primary objective is to learn a stochastic policy $\pi(a|s)$ to sample actions from an action space given a current state. In our setting, the action space represents possible locations where an additional edge can be placed to an existing graph and each state is represented by a current graph. Since under this training procedure we are sampling from the GFlowNet policy $P_F(s_i|s_{i-1}; \psi)$ at every iteration then attributing a reward associated to the sampled state/graph, the policy is susceptible to exploitation: if $P_F(s_i|s_{i-1}; \psi)$ samples a graph(s) with a high reward, it becomes easy for the policy to focus on sampling said graphs since they yield high reward. To alleviate this we encourage exploration using softmax tempering on our stochastic policy, by multiplying the logits of our forward policy by $1/c$ before applying the softmax function. A larger $c$ flattens the stochastic policy such that exploration within the action space is encouraged. However, setting the parameters $c$ is challenging and there exists a trade-off between exploring and exploiting the stochastic policy during optimization. We address this by using a cosine schedule for $c$ such that $1 \leq c \leq 1000$. We treat the period of the cosine schedule as a hyper-parameter.

## B.7  Discussion of Training Dynamics

GFlowNets are a relatively recent class of models that can be challenging to optimize. We discuss some of the challenges with training them especially in the context of a learned energy function.

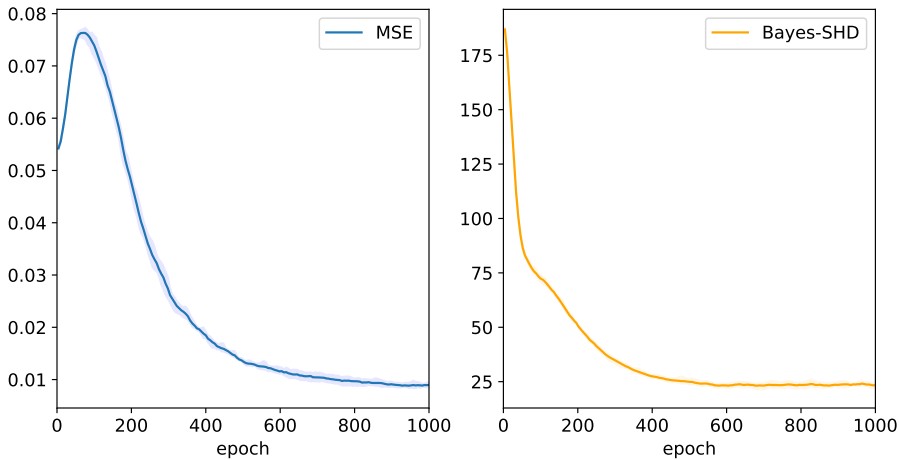

Figure 5: Validation curves throughout training of $l$-DynGFN on a linear system. Mean squared error (left) and Bayes-SHD (right).

We observed that in settings where the energy reward is fixed and we could proportionally penalize missing edges as well as the addition of incorrect edges (e.g. $\ell$-DynGFN), we were able to better learn posteriors over admissible graphs over models that require sparse priors and/or trainable energies. This suggests that DynGFN may be limited by an inadequate energy reward. However, we found training DynGFN with a trainable energy function challenging since the GFlowNet stochastic policy depends on the rewards, and vice versa. Further investigation and experimentation into this alternating optimization procedure is required. To in part alleviate the constraints of this challenge, in this work we presented results for $h$-DynGFN which used initialized GFlowNet parameters from a trained $\ell$-DynGFN model on the same task. We found that this initialization technique enabled more effective iterative optimization of the forward policy and energy reward. In Figure 5 we show validation curves throughout training of $\ell$-DynGFN on a linear system with $d = 20$ and a sparsity value of 0.9.

### B.8 Evaluating Quality of Learned Posteriors

Using our indeterminacy model defined in section 5, we can determine $P(G^*)$ for a given set of correlated variables, where $G^*$ denotes the set of true equally admissible structures. Here we assume $P(G^*)$ is a uniform distribution over $G^*$ and determine the KL between $Q(G)$ and $P(G^*)$. We compute the KL considering only the probabilities of a trained models to generate all structures in $G^*$, i.e. $Q(G^*)$. This is due to the computational constraints for calculating the KL for DynGFN since even for sparse graphs of moderate size this is a combinatorial computation. Nonetheless, this approach allows us to directly compare the learned posteriors $Q(G)$ to a ground truth discrete distribution over structure $G$ to evaluate the effectiveness of Bayesian structure learning approaches.

### B.9 Implementation Details

Our model is implemented in Pytorch and Pytorch Lightning and is available at `https://github.com/lazaratan/dyn-gfn`. Models were trained on a heterogeneous mix of HPC clusters for a total of ~1,000 GPU hours primarily on NVIDIA RTX8000 GPUs.

# C  Additional Details

## C.1  Single-cell Biology and Gene Regulatory Network Inference

### C.1.1  Gene Regulatory Networks and Cell Dynamics

One dynamical system of interest is that of cells. Cellular response to environmental stimuli or genetic perturbations can be modelled as a complex time-varying dynamical system [17, 1]. In general, dynamical system models are a useful tool for downstream scientific reasoning. In this work we are primarily interested in identifying the underlying cell dynamics from data. A reasonable model for cell dynamics is as a stochastic dynamical system with many, possibly unobserved, components. There are many data collection models for gaining insight into this system from single-cell RNA-sequencing data. We will primarily focus on RNA velocity type methods, where both $x$ and an estimate of $dx$ are available in each cell, but note that there are other assumptions to infer dynamics and regulation such as pseudotime-based methods [47, 1], and optimal transport methods [17, 48, 51, 22, 23]. After learning a possible explanatory regulation, this is used in downstream tasks, but the resulting conclusions drawn from these models are necessarily conditional on the inferred regulation. Motivated by gene regulatory networks, we explicitly model uncertainty over graphs which allows us to propagate the resulting uncertainty to downstream conclusions.

### C.1.2  Learning Gene Regulatory Networks From Single-cell Data

Single-cell transcriptomics has an interesting property in that from a single measurement we can estimate both the current state $x$ and the current velocity $dx$. Because mechanistically RNA undergoes a splicing process, we can measure the quantities of both the unspliced (early) and spliced (late) RNA in the cell. From these two quantities we can estimate the current RNA content for each gene and the current transcription rate. There exist many models for denoising and interpreting this data [27, 9, 45]. Furthermore, there exist more elaborate measurement techniques to extract more accurate velocity estimates [45]. The fact that we have an estimate of the current velocity is exceptionally useful for continuous time structure discovery because it allows us to avoid explicitly unrolling the dynamical system.

Learning the underlying causal structure from data is one of the open problems in biology. There are many works that attempt to learn the effect of a change in a gene, or the addition of a drug. These works often build models that directly predict the outcome of an intervention. This may be useful for certain applications, but often does not generalize well out of distribution. We would like to learn a model of the underlying instantaneous dynamics that give rise to effects at longer time scales. This approach has a number of advantages. (1) it is closer to the mechanistic model; it may be easier to learn a model of the instantaneous dynamics rather than the dynamics over long time scales (details in Appendix C.2). (2) One model can be trained and applied to data from many sources including RNA-velocity, Pseudotime, Single-cell time series, and steady state perturbational data. (3) The instantaneous graph may be significantly sparser (and therefore easier to learn) than the summary graph or the equilibrium graph.

### C.1.3  Further Details on Experiments with Single-cell RNA-velocity Data

The process of Eukaryotic cell division can be divided into four well regulated stages based on the phenotype, Gap 1 ($G_1$), Synthesis ($S$), Gap 2 ($G_2$), and Mitosis ($M$). This process is a good starting point for GRN discovery as it is (1) relatively well understood, (2) deterministic, and (3) well studied with plentiful data. While there is an underlying control loop controlling the progression of the cell cycle, there are many other genes that also change during this cycle. To rediscover the true control process from data we must disentangle the true causal genes from the downstream correlated genes. This may become very difficult when we only observe dynamics at longer time scales. We hide a cell cycle regulator among two downstream genes that are highly correlated (Spearman $\rho > 0.75$) and test whether we can model the Bayesian posterior – namely that we are uncertain about which of the three genes (Cdc25A, Mcm2, or Mcm5) is the true causal parent of Cdk1.

## C.2 Instantaneous Graph and Long-horizon Graph

The graph recovered depends on the time scale considered. We make a distinction between the conditional structure of the graph based on the time scale. Consider a system governed by $\frac{dx(t)}{dt} = f(x(t))$. We define the instantaneous graph as:

$$g_{i,j} := \cup_x \mathbf{1}\left(\frac{df(x)_j}{dx_i}\right), \qquad i = 1, \ldots, d, j = 1, \ldots, d. \tag{6}$$

Here, $\mathbf{1}(z) = 1$ if $z \neq 0$, otherwise $\mathbf{1}(z) = 0$. We can define the temporal summary of the system drift $f$ as:

$$F(T, x) = x_0 + \int_0^T f(x(t))dt. \tag{7}$$

Then, we can define a long-horizon graph over long time-scales as:

$$G_{i,j} := \cup_x \mathbf{1}\left(\frac{\partial F(T, x)_j}{\partial x_i}\right), \qquad i = 1, \ldots, d, j = 1, \ldots, d. \tag{8}$$

If we are integrating the drift $f$ over long time-scales, the long-horizon graph may be less sparse than the instantaneous graph. In cellular systems, this equates to observing cell dynamics over long time-scales, in turn observing increase quantities of correlations between variables of the system. Thus, trying to delineate the instantaneous dynamics from long time-scales may be difficult depending on the underlying system dynamics.

## C.3 Further Discussion on Future Work

Although DynGFN is currently limited to smaller systems, we foresee approaches that would enable some degree of scaling DynGFN to larger systems. One approach is to leverage biological information of known gene-gene connections as a more informative prior for DynGFN. Currently, DynGFN learns a forward stochastic policy for $Q(G|D)$ starting from an initialized state $s_0$ of all zeros. Instead, we can define $\tilde{s}_0$ using a prior of high confidence biological connections and sequentially add edges starting from this new initial state. This would reduce the number of possible structures DynGFN would need to search over, thus improving the potentially scalability of DynGFN. Another approach is to learn structure between sets of genes (variables) rather than single genes. Since GRNs are generally very sparse, it makes sense to group genes in sets. Consequently, we can then learn structure between these grouped genes, rather than just individual genes. In turn, DynGFN can explore/learn the structure over a smaller space while effectively capturing structure between a significantly larger set of genes. To group genes, we would use prior biological information, either form existing literature or expert domain knowledge.

In this work we exploit the use of a minimal prior, i.e. $L^0$ sparsity prior, for learning Bayesian dynamic structure between variables. In general, the aforementioned approaches for scaling DynGFN to larger systems involve the use of more informative priors on $G$. Although we mention two ways we foresee approaching this in the biological context of GRN inference, the general approach of using more informative priors can help scale DynGFN to larger systems across applications.

## C.4 Broader Impacts

While it is important to acknowledge the potential risks of drawing incorrect scientific conclusions due to incorrect assumptions, our work embraces a Bayesian perspective for structure learning. A key component of our work is to account for uncertainty within our method, aiming to minimize the chances of incorrect conclusions. It is important to note that the accuracy of conclusions relies on applying the method in settings that align with the underlying assumptions, such as causal sufficiency and the use of dynamic observational data. By adhering to these guidelines, our approach holds promise for producing robust and reliable scientific outcomes.

