# OpenReview forum: "DynGFN: Towards Bayesian Inference of Gene Regulatory Networks with GFlowNets"
_NeurIPS.cc/2023/Conference — NeurIPS 2023 poster_

### Official Review · Reviewer_2wco · 2023-07-05

**Soundness:** 3 good
**Presentation:** 2 fair
**Contribution:** 3 good
**Rating:** 6
**Confidence:** 3

**Summary:**

In this work, the authors propose an extension of GFlowNets to enable posterior sampling of cyclic graph structures and apply their approach to infer gene regulatory networks (GRN). While existing methods are able to infer cyclic graph structures or sample from Bayesian posteriors over DAGS, they cannot do both simultaneously. The authors formulate the problem as a dynamical systems identification problem which can be factored into the priors and model likelihood. The priors are learned using either linear or non-linear Hyper Networks while the model likelihood is estimated using the GFlowNets framework with detailed balance loss. The authors show their model is robust to several parameters including edge sparsity and time intervals between data points. Furthermore, they introduce several optimization ideas enabling a tractable search over admissible graph structures. Finally, they benchmark their approach against several baseline methods and apply their approach to an RNA velocity dataset to show their method is able to recover known and putative gene-gene interactions.


**Strengths:**

The paper addresses an interesting/important question which has been the center of much research for decades: posterior sampling over directed graphs, this time with cycles and not DAGs. The authors appear knowledgable in this field and they take advantage of GFlowNets, building a method to assess posterior structures with cycles as they unfold them across time.

**Weaknesses:**

Our biggest grievance is that the paper is just not reader friendly for anyone who is not an expert in that specific line of work.
As noted the work seems technically very solid, one that offers specific new solution for the stated problem. Still, we as researchers not in this specific field, remain unsure after reading it of the overall innovation/significance of creating posterior sampling of structures over GFlowNets. We add more details below.

(1)
The termination criteria of Algorithm 1 is not clear. Under what conditions does the transition probability become null? Although one of the main innovations of the paper is sampling from the posterior of GFlowNet structures, the authors don’t reference any standard Bayesian analysis. How are posterior samples summarized (for example, how is a point estimate of p obtained from posterior samples in Figure 3)? How and when does Algorithm 1 converge to the posterior? Is the approach sensitive to variable initializations?

(2)
Although the authors compare their approach to sensible baselines using sensible metrics, it is not clear the approach has a clear advantage over the original GFlowNet in this applied setting. For example, how does the MAP parameter estimate (or equivalent)  of this method compare to the original GFlowNet under the AUC metric (these are presumably comparable). Does GFlowNet produce different insight on gene-gene interactions in the RNA velocity dataset?

(3)
The authors assume causal sufficiency as a criteria for model identifiability. Is this a reasonable assumption given the sparsity of single cell data and limitations in the scalability of the model? In other words, how can we be reasonably certain that all casual gene measurements are observed given the model effectively only functions on a gene space of size 20 or lower?

(4)
Section 5 on graph augmentation is very abstruse. Efforts could be made to improve readability/accessibility for an audience that doesn’t work directly in this field. The related Figure 2 is also very cryptic.

(5)
It is not clear or obvious how the factorization into the “per-node” posterior in section 4.1 leads to the reduction in search space from 2^d^2 to d2^d. A brief explanation would be useful.

(6)
Line 154: “Previous work has shown GFlowNets are useful in settings with multi-modal posteriors.” This needs a citation.

(7)
There is inconsistency in the notation in Figure 1 which leads to some confusion. Specifically, it appears the Q variable is overloaded. How is the graph prior (represented as an adjacency matrix) sampled? Is there a hyperprior that is used or are graph structures uniform?

(8)
The patterns that the authors suggest that are evident in the Figure 3 heatmaps are not convincing. Perhaps it would be better to sort both heatmaps the same way instead of independent clustering.

**Questions:**

see above.

**Limitations:**

see above.

---

> ### Author Rebuttal · Authors · 2023-08-10
>
> We thank the reviewer for their comments, insightful feedback, and helpful suggestions to improve the manuscript. Please refer to our feedback summary and the attached document regarding questions/clarifications on Fig. 1, Section 5 and Fig. 2, and Fig. 3. In the following, we address individual comments/questions brought up by the reviewer to improve the overall exposition quality of the manuscript. We are happy to incorporate any additional suggestions and feedback.
>
> >The termination criteria of Algorithm 1 is not clear. How and when does Algorithm 1 converge to the posterior?
>
> While the stopping criteria is not explicit, it is part of the GFlowNet policy $P_F(s_i | s_{i-1}; \psi)$. See [1] for further details and for sufficient conditions under which the optimal GFlowNet policy (which includes the stopping criterion) correctly converges to the posterior.
>
> >How are posterior samples summarized? How is a point estimate of p obtained from posterior samples in Fig. 3?
>
> We are not certain which $p$ is being referred to, but any statistics of the posterior are computed by either Monte Carlo (sampling from the GFlowNet policy) or using log-likelihood evaluation of the model. In Fig 3 there is a $\rho$ which quantifies correlation between genes used to determine the ground truth graph. Is this what is being referred to?
>
> >Is the approach sensitive to variable initialization?
>
> We do not think DynGFN is very sensitive to variable initialization. Examining Tables 1-4 in Section 6 we see that the standard deviation over 5 seeds is much smaller than the differences between methods. This suggests that DynGFN is relatively stable in that it outputs similar performing posteriors over different initializations.
>
> >Section 5 on graph augmentation is very abstruse.
>
> To clarify this section, we have amended Fig. 2 (see attached document) and included more informative labels and captions to correspond to the language used in Section 5.
>
> >It is not clear that the approach has a clear advantage over the original GFlowNet in the applied setting?
>
> A key contribution of this paper is on how to leverage time and velocity information to address Bayesian structure learning over **cyclic** graphs representing dynamics, while DAG-GFN [2] (we assume this is what is meant by original GFlowNet) only works in the acyclic case. Therefore, it is not directly applicable to graphs with cycles, as are known to frequently occur in gene regulatory networks. We refer the reviewer to Fig. 4 in the attached PDF where we show an explicit example where using the DAG assumption for cyclic structure learning fails.
>
>
> >Is the causal sufficiency assumption a reasonable assumption..?
>
> The reviewer identifies a great point. We agree that the assumption of causal sufficiency likely fails for single-cell data with only 20 genes. However, as we grow the number of genes that we are able to infer over, this assumption becomes more plausible. Ideally, we want an approach that yields (1) good posteriors, (2) good identifiability, and (3) good scalability, but due to the significant challenges associated with each of them, we have devoted our effort to address the first two in this work. If GFlowNets (or some other inference procedure) improved scalability, DynGFN could immediately leverage this advancement to do inference over a larger number of genes, which would make the assumption more plausible.
>
> >It is not clear or obvious how the factorization into the “per-node” posterior in section 4.1 leads to the reduction in search space.
>
> The per-node factorization in effect uses $d$ GFlowNets to learn $d$ graphs each of size $(d \times 1)$ for $dx_i$ and its parents $x$ (ref equation 4). This is possible because the posterior is factorizable into independent components. This is an advantage of DAG-GFN [2] as this is not possible in the DAG setting. Each of the d graphs are then aggregated to form the full $(d \times d)$ graph $G$. In this case, each of the $d$ GFlowNets trains over a search space of $2^d$ combinations. Therefore, the state space of the per-node factorized DynGFN is $d \cdot 2^d$. In contrast, if a single GFlowNet is used, then there are $2^{(d^2)}$ possible states. We will include this explanation in section 4.1. Additionally, we thank the reviewer for noting this and bringing it to our attention. We correct line 191: $d \cdot 2^d$ = $20 \cdot 2^{20} ~= 2^{104}$ for $d=20$ should be $d \cdot 2^d ~= 2^{(4.3 + 20)}$ for $d=20$.
>
> >Previous work has shown GFlowNets are useful in settings with multi-modal posteriors. This needs a citation.
>
> We thank the reviewer for noting this. We agree, we will add citations [1-4] to this statement.
>
> >Figure 3
>
> See the attached PDF for an updated Figure 3 with the same ordering. To make our point more obvious, we additionally include a histogram of correlation values for the full correlation and the correlation over cell cycle time. We can see that the (absolute value) of the correlation over cell cycle time is substantially higher on average in distribution.
>
> We thank the reviewer for their insightful and helpful feedback on our paper. We hope that our rebuttal fully addresses all the important points raised by the reviewer. If our responses, alongside the accompanying additions, improve the overall quality of our work, we hope the reviewer would kindly raise their score. We are more than happy to answer any further questions.
>
> [1] Bengio, Yoshua, et al. "Gflownet foundations." arXiv preprint arXiv:2111.09266 (2021).
>
> [2] Deleu, Tristan, et al. "Bayesian structure learning with generative flow networks." Uncertainty in Artificial Intelligence. PMLR, 2022)
>
> [3] Malkin, Nikolay, et al. "Trajectory balance: Improved credit assignment in gflownets." Advances in Neural Information Processing Systems 35 (2022)
>
> [4] Madan, Kanika, et al. "Learning GFlowNets from partial episodes for improved convergence and stability." International Conference on Machine Learning. PMLR (2023)

---

> > ### Comment · Reviewer_2wco · 2023-08-17
> > **Feedback for the rebuttal**
> >
> > We have read through the authors response and appreciate the effort made to fix/clarify points we raised. As noted, the "causal sufficiency assumption" is a very strong assumption and as the authors admit with current scale (~20 genes) far from realistic. This limits the actual applicability of the work and makes it more of a computational/methodological contribution. Nonetheless, we appreciate the work/contribution and retain our (positive) score.

---

> ### Comment · Area_Chair_6Upb · 2023-08-17
> **what did you think of the authors' response?**
>
> The authors have provided detailed responses to your questions and comments. Please revise the text and score of your review to reflect how their responses have changed your perspective on their submission, and please acknowledge that you have read the authors' carefully written response.

---

### Official Review · Reviewer_uDCE · 2023-07-06

**Soundness:** 2 fair
**Presentation:** 2 fair
**Contribution:** 3 good
**Rating:** 6
**Confidence:** 3

**Summary:**

The authors tackle the well-known problem of Gene Regulatory Network (GRN) inference from time series data. Starting from an estimate of RNA velocity, GRN inference is framed as a causal discovery task, with two specificities: the inferred graphs may be cyclic, and a probabilistic distribution over candidate graphs is sought rather than a point estimate, so as to account for the high amounts of noise inherently present in biological data. The method is articulated around two assumptions:

- a dynamic structural model $\frac{dx_i(t)}{dt} = f_i(\text{Pa}(x_i), \epsilon_i)$ for each gene. $\text{Pa}(x_i)$ is directly obtained from the regulatory graph $G$, and the structural causal model (SCM) $f_i$ is parametrized by $\theta$
- a factorisation of the joint distribution of the graph $G$, the structural causal model parameters $\theta$, and the observed data $\mathcal{D}$ as $p(G,\theta,\mathcal{D}) = p(\mathcal{D}|G,\theta)p(\theta|G)p(G)$. The graph sampler $p(G)$ is modeled as a GFlowNets. Next, since the parameters of the SCM depend on the graph structure, they are computed by a HyperNetwork, i.e. $p(\theta|G)$ is substituted to a neural network which takes as input a graph $G$ and outputs the structural equation model parameters $\theta$: $p(\theta|G) = \delta(\theta|G)$.

Next, the authors evaluate the proposed method, DynGFN, on several synthetic examples as well as a real-world one involving single-cell RNA velocity data. The authors introduce other baselines by using different approaches than a GFlowNets for the graph sampler $p(G)$. This being said, DynGFN offers substantial improvements when it comes to jointly recovering the ground truth structure while characterizing the uncertainty around it.

**Strengths:**

The big picture of the method is well explained. The work described here is likely to have a significant impact for the GRN inference community, as I believe that not many probabilistic methods are available in this field. I also believe that DynGFN could be used as a starting point for the more general problem of physical/biological dynamical system discovery, where one is not only concerned with the graph adjacency matrix but with providing a mechanistic description of the structural causal model $f_i$ [1,2]

[1] Discovering governing equations from data by sparse identification of nonlinear dynamical systems, PNAS, 2016
[2] Identification of dynamic mass-action biochemical reaction networks using sparse Bayesian methods, PLoS Comp. Biol., 2022

**Weaknesses:**

- The paper builds on GFlowNets for the graph sampler $P(G)$ and Hypernetworks to generate the structural causal model parameters $\theta$ given a sampled graph $G$. I found it slightly difficult to understand everything $in$ $detail$ given that these tools are each introduced only in a small paragraph.

- Even though the method focuses on Bayesian Inference, it would have been interesting to assess DynGFN's performances compared to SOTA methods for GRN inference from time series data such as DynGENIE3 [1], BINGO [2] or others. Lastly, these methods use time-resolved gene expression data, from which one can get an estimate of the RNA velocity using for instance finite differences, which may be a quite rough estimate for sparsely sampled measurements. Do I understand right that the RNA velocity data employed in this submission somehow constitutes a ``more principled'' way to estimate the RNA velocity?

[1] dynGENIE3: dynamical GENIE3 for the inference of gene networks from time series expression data, Scientific Reports, 2018
[2] Gene regulatory network inference from sparsely sampled noisy data, Nature Communications, 2020

**Questions:**

- Any insights on why $h$-DynGFN yields significantly better results than $\ell$-DynGFN on the linear system presented in table 1?

-  In the single cell experiment, where does the ground truth network (used to compute AUC / Bayes-SHD) come from? Could you explain a little more what does "correlation over cell cycle time" means in Section 6.3?

I have read the rebuttal done by the authors.

**Limitations:**

The authors mention as main limitation the difficulties to scale to large gene regulatory networks due to the combinatorial explosion. Causal sufficiency is also assumed here, which means that all relevant variables are observed. This assumption hardly ever holds in real-world biological problems. This issue is also mentioned by the authors in the Appendix.

---

> ### Author Rebuttal · Authors · 2023-08-10
>
> We thank the reviewer for their comments and insightful feedback. We address your comments in what follows.
>
> >The paper builds on GFlowNets for the graph sampler and Hypernetworks to generate the structural causal model parameters. I found it slightly difficult to understand everything given that these tools are each introduced only in a small paragraph.
>
> Thank you for bringing up this issue, we realize our work is quite dense and the page limit was an issue. We will of course expand the exposition in the main text with the addition of an extra page. We are happy to incorporate any specific suggestions you have on how to improve the exposition.
>
> > Even though the method focuses on Bayesian Inference, it would have been interesting to assess DynGFNs performance to SOTA methods from time-series data.
>
> We are also very interested in how DynGFN compares to and can be combined with SOTA methods on inferring GRNs from time-series data! As noted by the reviewer, methods such as DynGENIE3 and BINGO consider a different setting where we have access to multiple samples from the time series. They are generally not well suited to velocity measurements, which effectively amounts to having two extremely close time points, as these methods often using changes over time to infer the gene regulatory network dynamics. We believe adapting these state of the art methods to infer regulation from velocity data would be an interesting direction given the potential quantity of RNA-velocity analysis, but we believe this is out of scope for this project. Future work on how to adapt DynGFN to the time series setting, potentially using ideas developed in [3], is also extremely promising.
>
> >Does the RNA velocity used in this submission constitute a more ‘principled’ way of estimating RNA velocity?
>
> We do not explicitly estimate RNA velocity in this work. Rather, we use estimated RNA velocity (using scVelo [1]) to help formulate the problem of Bayesian structure learning over cyclic graphs using observational data consisting of dynamic tuple pairs $(x, dx)$. We leave inference of RNA velocity using DynGFN for future work. See response to Reviewer [b9zX] for further discussion.
>
> >Insights on why h-DynGFN yields better results compared to l-DynGFN on the linear system?
>
> The reviewer brings up an insightful question. We suspect that l-DynGFN may be more sensitive to the stochasticity between batches of data compared to h-DynGFN since the analytic linear solver used in l-DynGFN directly uses the minibatch to solve for the parameters (the factorization P(\theta | G) is poorly enforced). This means that there is greater stochasticity in the reward depending on the batch selected. We believe this is why we see that l-DynGFN yielding worse results on the linear system compared to h-DynGFN.
>
> >Where does the ground truth network in the single-cell experiment come from?
>
> This ground truth network is constructed using external prior biological knowledge. Specifically, we extracted a subset of the gene network from the KEGG cell cycle pathway entry hsa04110.
>
> >What is meant by ‘correlation over cell cycle time’?
>
> This data is from Riba et al. 2022 [2]. Here they supply public data with a cell cycle pseudo-time label called `cell_cycle_theta` in the public data. We correlate bin this label into 10 bins and approximate the correlation of gene expression over time.
>
> We would like to thank the reviewer for their review of our paper. We believe we have answered all the great points raised by the reviewer in our author response, and we kindly ask the reviewer to consider upgrading their score if the reviewer is satisfied with our responses. Please let us know if you have any additional feedback or comments. We would be happy to discuss.
>
> [1] Bergen, Volker, et al. "Generalizing RNA velocity to transient cell states through dynamical modeling." Nature biotechnology 38.12 (2020)
>
> [2] Riba, Andrea, et al. "Cell cycle gene regulation dynamics revealed by RNA velocity and deep-learning" Nature Communications (2022)
>
> [3] Tong et al. "Simulation-Free Schrödinger Bridges via Score and Flow Matching" ArXiv (2023)

---

> ### Comment · Area_Chair_6Upb · 2023-08-17
> **what did you think of the authors' response?**
>
> The authors have provided detailed responses to your questions and comments. Please revise the text and score of your review to reflect how their responses have changed your perspective on their submission, and please acknowledge that you have read the authors' carefully written response.

---

### Official Review · Reviewer_b9zX · 2023-07-07

**Soundness:** 4 excellent
**Presentation:** 2 fair
**Contribution:** 3 good
**Rating:** 6
**Confidence:** 4

**Summary:**

Authors introduced a principled methodology called DynGFN, which effectively identified cyclic structures and concurrently modeled the Bayesian posteriors over directed acyclic graphs (DAGs). Leveraging RNA velocity, the authors formulated a dynamic system that unveiled the underlying gene regulatory networks. DynGFN was meticulously designed with three modules, and its superior performance was demonstrated through synthetic experiments and real-world analysis.

**Strengths:**

The real biological systems can be rarely formulated as DAGs and there are always feedback loops to make the system work. DynGFN was appropriately motivated with a real biological thinking.

**Weaknesses:**

The overall manuscript was hard to follow as there was too much content included. There were still a lot of technical details that needed to be clarified. It would improve substantially if the manuscript can be carefully restructured.

**Questions:**

Major concerns
1. The synthetic experiments and real analysis did not explicitly present any results related to feedback loop identification, which serves as one of the motivations behind DynGFN. Could you elaborate on this point?
2. How many epochs did it take to stabilize the training of l-DynGFN?
3. Given the inference of RNA velocity involves multiple free parameters, how do these parameters impact the inference results obtained from DynGFN?
4. In the synthetic experiments simulating dynamic systems, how were the ground truth regulatory networks constructed based on the simulated dynamic systems?
5. Appendix B.3 mentions the construction of the validation and test sets to fine-tune the hyperparameters. Could you explain how these sets were constructed for the experiment?
6. Regarding Table 4, could you clarify what the ground truth graph was used for calculating Bayes-SHD and AUC? Was it formulated based on external knowledge?
7. In Figure 3, it is important to note that gene regulation is not solely determined by the direct interaction between corresponding proteins. In fact, it has been reported that "CDK1 targets MCM2-7 complex..." [1].

[1] Enserink, Jorrit M., and Richard D. Kolodner. "An overview of Cdk1-controlled targets and processes." Cell division 5.1 (2010): 1-41.



Minor concerns
1. Figure 1 lacks definitions for numerous parameters, which may cause confusion in understanding the illustration.

**Limitations:**

The authors mentioned two limitations: 1) scalability issues concerning larger systems and 2) hyperparameter tuning. To address these challenges, the authors suggested employing more informative priors or grouping genes together as nodes.

---

> ### Author Rebuttal · Authors · 2023-08-10
>
> We thank the reviewer for their comments and insightful feedback. Please refer to our summary to all reviewers and the attached document regarding questions about cyclic dependencies in our synthetic and real experiments, as well as for clarifications regarding Fig. 1. We also introduced a new toy example to exemplify our model’s capability for learning cyclic dependencies (See attached Fig. 4 and response to Reviewer [PVQy]). We now address each salient point individually.
>
> >How many epochs for l-DynGFN to stabilize?
>
> To show when l-DynGFN stabilizes during training, we have introduced validation curves for mean squared error and Bayes-SHD over the course of training on the linear system with $d=20$ (See Fig. 5 in the attached). In this experiment I-DynGFN stabilizes in around 500 epochs.
>
> >How do RNA velocity parameters impact the inference results obtained by DynGFN?
>
> The reviewer points at a great question. We agree that these free parameters may impact the results of DynGFN. DynGFN currently takes RNA velocity as input, but an ideal solution would incorporate this source of uncertainty (uncertainty due to RNA velocity inference) into the posterior over the graph and the parameters. Due to the difficulty of the cyclic structure learning problem and given that DynGFN already requires multiple posteriors, we leave investigation into a fully Bayesian RNA velocity and gene network recovery method to future work. We will add discussion of this to the limitations section.
>
> >How were the ground truth regulatory networks constructed for the synthetic dynamical systems?
>
> We first construct the ground truth regulatory network (GRN) and then use this ground truth GRN to simulate the system dynamics. Consider the linear system $dx = Ax$. To construct the system, we randomly sample $A$ such that $A \in \mathbb{R}^{d \times d}$ with a specified sparsity. We constrain $A$ to follow some properties to ensure the system is stable. Specifically, we subtract the maximum eigenvalues from the diagonal of $A$. This $A$ is then used to simulate $dx = Ax$. The procedure follows for the non-linear system with the addition of the non-linearity. We will include this description in the Appendix. For more specific implementation details see line 58 of `src/datamodules/simulated_datamodule.py` in the attached code.
>
> >How were the validation and test sets constructed for the experiments? How was the ground truth graph in the real data setting formulated?
>
> In the synthetic experiments, we simulate a system and generate train, validation, and test observations $(x, dx)$ for a given system with true connections defined by $A$. In the real data setting, we split the observed data $(x, dx)$ into train, validation, and test datasets. $A$ in the real data setting is constructed using prior biological knowledge. Specifically, we extracted a subset of the gene network from the KEGG cell cycle pathway entry hsa04110. Fig. 3 (see attached document for updated version) shows the constructed ground truth graph.
>
> >It is important to note that gene regulation is not solely determined by the direct interaction between corresponding proteins.
>
> We agree, we will add a mention of this in Section 6.3, and to the future work section. We find this quite interesting and gets back to what gene regulatory network we are trying to discover. Our model is searching for a posterior over sparse gene regulatory networks which explain the data dynamics. It could be argued that as the time goes to zero, we will find only direct interactions, however we leave further investigation of this to future work.
>
> We thank the reviewer for their time and effort in reviewing our work and we hope the reviewer would kindly consider a fresh evaluation of our work given the main clarifying points outlined above. If you have any additional comments, we would be happy to discuss further.

---

> > ### Comment · Reviewer_b9zX · 2023-08-21
> >
> > I have read through the responses from authors and I appreciate the efforts to further improve the manuscript. NOTEARS is inherently not a suitable method to identify cyclic relations in the data. An alternative approach will be preferred. Therefore I keep my original score unchanged.

---

> ### Comment · Area_Chair_6Upb · 2023-08-17
> **what did you think of the authors' response?**
>
> The authors have provided detailed responses to your questions and comments. Please revise the text and score of your review to reflect how their responses have changed your perspective on their submission, and please acknowledge that you have read the authors' carefully written response.

---

### Official Review · Reviewer_PVQy · 2023-07-24

**Soundness:** 2 fair
**Presentation:** 2 fair
**Contribution:** 2 fair
**Rating:** 7
**Confidence:** 4

**Summary:**

The authors propose to learn dynamical systems with cyclic dependencies. They factorise the generative model using a variant of the GFlowNet model, a HyperNetwork and MLPs.

**Strengths:**

Positives include:
1. extending the formulation from DAGs to cyclic graphs
2. introducing a per-node posterior formulation  to improve the computational complexity
3. using GFlowNets to address the multimodality issues

**Weaknesses:**

Negatives include:
1. due to the high density the paper is hard to understand: a lot is explained and detailed in the Appendix
2. it is hard to gauge the quality of the results: even for the artificial case it is hard to assess intuitively if the approach solves the problem in a satisfactory way. Perhaps the authors should use simple systems with few nodes, with and without cycles, and show how well the true dependencies are recovered.
3. it is unclear how important it is to being able to model cyclic structures: where are the cyclic dependencies in the examples shown in Fig. 2 and Fig. 3? The authors should show a clear example where failing to model the cyclic dependencies results in a poor fit.
4. it is unclear how to translate the declared advantage of being able to model multi-modal posteriors  in practical terms: how would we make use of this representation capacity when explaining the cause dependencies in a gene regulatory network?

**Questions:**

To illustrate the advantages of the proposed approach:
1.   use simple systems with few nodes, with and without cycles, and show how well the true dependencies are recovered.
2. show a clear example where failing to model the cyclic dependencies results in a poor fit.
3. show how to use the multimodal representation capacity when explaining the cause dependencies in a gene regulatory network.

**Limitations:**

Yes.

---

> ### Author Rebuttal · Authors · 2023-08-10
>
> We thank the reviewer for their comments and insightful feedback. Please refer to our feedback summary and the attached document regarding questions about cyclic dependencies in Fig. 2 and Fig. 3. We also include further discussion regarding the toy example in the feedback summary. We now address each comment individually.
>
> >Perhaps the authors should use simple systems with few nodes, with and without cycles, and show how well the true dependencies are recovered.
>
> The reviewer suggests a valuable addition. To convey this, we introduced a toy example with a simple 3 variable system (with and without cycles), see Fig. 4 in the attached document. We consider a method for learning acyclic graphs (NOTEARS [1]) which does not model cyclic dependencies, unlike DynGFN. It is clear from this example that NOTEARS struggles to recover cyclic dependencies. We will add Fig. 4 to the manuscript. We can also verify this result by considering a conditional independence test over cyclic dependencies. With reference the attached Fig. 4 it is easy to see that the conditional independence test fails in the cyclic setting: In the acyclic case, we can identify the v-structure by observing that $x_1 \perp x_3$ and $x_1 \not\perp x_3 | x_2$, which implies that $x_2$ is a collider (i.e. $x_1$ and $x_3$ are marginally independent and conditionally dependent); in the cyclic example, we introduce time dependencies such that there are cycles in the summary graph that render these variables marginally dependent.
>
> >it is unclear how to translate the declared advantage of being able to model multi-modal posteriors in practical terms
>
> Learning multi-modal posteriors over structure allows us to quantify uncertainty over a particular causal dependency. In turn, it makes it easier to answer the active learning question, “how should we select interventions such that we minimize uncertainty?” [2]. We use this as motivation for our work and cite relevant papers (see lines 54-55). We will add the aforementioned citation to this list. As a first step, we need to effectively learn the multi-modal posterior over structure, hence the motivation and focus of this work.
>
> We hope our response fully addresses all the important and salient points raised by the reviewer. We believe the new example improves the clarity of our work, and we thank the reviewer for their excellent suggestion. Due to the increased clarity of these new examples, we ask that the reviewer consider raising their score. If you have any additional comments and suggestions we would be happy to discuss further.
>
> [1] Zheng, Xun, et al. "Dags with no tears: Continuous optimization for structure learning." Advances in neural information processing systems 31 (2018)
>
> [2] Toth, Christian, et al. "Active bayesian causal inference." Advances in Neural Information Processing Systems 35 (2022)

---

> ### Comment · Area_Chair_6Upb · 2023-08-17
> **what did you think of the authors' response?**
>
> The authors have provided some responses to your questions and comments. Please revise the text and score of your review to reflect how their responses have changed your perspective on their submission, and please acknowledge that you have read the authors' carefully written response.

---

> > ### Comment · Reviewer_PVQy · 2023-08-18
> >
> > The authors have provided a satisfactory answer to the main concerns we raised. The high density and not being a self contained  paper is still an issue. I will therefore correspondently raise the score.

---

### Author Rebuttal · Authors · 2023-08-10

We would like to thank all the reviewers for their time and insightful feedback when reviewing our paper. We appreciate the constructive criticisms and suggestions which will serve to improve the overall quality of our paper.

Our method focuses on Bayesian structure learning over cyclic graphs for application in gene regulatory network (GRN) inference. We consider a dynamical systems perspective to model cycles through time while leveraging advancements in generative flow networks (GFlowNets) to learn complex posteriors over explanatory structure given the observational data. In general, reviewers found our work to address an important and long-standing problem in biology while employing a probabilistic/Bayesian perspective ([uDCE], [2wco]), to be well motivated for tackling the problem of cyclic Bayesian structure learning for GRNs ([PVQy, b9zX, uDCE]), and to provide a good quality contribution to the area ([b9zX, uDCE, 2wco]). Reviewers’ primary concerns consist of clarifying questions. Here we address some general concerns raised across reviewers.

1. **Cyclic dependencies in synthetic and real experiments:** Reviewers asked about cyclic dependencies in the synthetic and real experiments ([PVQy], [b9zX]). We highlight that our experiments do all contain cyclic dependencies in the ground truth graphs and amend the accompanying figures (Fig. 2 and Fig. 3) to appropriately demonstrate this. Please see the attached document for reference. In Fig. 2 we have added a 3D realization showing the difference between the dynamic graph and static graph. We show how we model cyclic dependencies present in the static graph through directed edges in the dynamic graph. This corresponds to the motivation of using dynamic data of the form (x, dx). In Fig. 3 we have adjusted the presented GRN to include the cyclic dependencies of the ground truth GRN that were considered in the experiment.
2. **Comparison to DAG learning method(s):** Reviewers raised points about demonstrating the effectiveness of DynGFN to learn cyclic dependencies relative to counterpart directed acyclic graph (DAG) learning methods ([PVQy], [2wco]). To show this, we have added an additional experiment (toy example) that considers a DAG-based system and accompanying system with cyclic dependencies. We demonstrate that the DAG structure learning method (NOTEARS [1]) cannot learn the cyclic dependencies in the cyclic system, compared to DynGFN which performs very well in the cyclic setting. We show this result in Fig. 4 (included in the attached document) which we will include in the revised manuscript.
3. **Figure clarifications:** Some reviewers asked clarifying questions regarding presentation of Fig.1 ([b9zX], [2wco]) and Fig. 3 ([PVQy], [b9zX], [uDCE], [2wco]). To clarify Fig. 1, we will add subscripts to the $Q$ variable to separate usage for modeling the posterior over graphs and posterior over parameters to outline the difference between the 2 $Q$’s. Specifically, for posterior over graphs we will state $Q_\Psi(G | D)$ (this is consistent with the notation in the figure), while for the posterior over parameters we will state $Q_\phi(\theta | G, D)$. Since we model $Q_\phi(\theta | G, D)$ as a Dirac, this collapses to $\theta = h_\phi(G)$, as shown in the figure. For Fig. 3, we have amended the ground truth GRN (stated in item 1 above) and the accompanying heatmaps. Please see the attached document for the amended Fig.3.

[1] Zheng, Xun, et al. "Dags with no tears: Continuous optimization for structure learning." Advances in neural information processing systems 31 (2018)

---

### Comment · Area_Chair_6Upb · 2023-08-17
**acknowledge and discuss the authors' detailed response**

The authors have provided a detailed response, which includes answers to reviewers' questions and more exposition of the technical material. **Drop a comment in this thread with how their response changed your perspective on the submission.**

Remember to also update your scores/reviews to reflect how your views have changed in light of the authors' response, and respond to comments/questions the authors have left underneath individual reviews.

---

### Decision · Program_Chairs · 2023-09-21

**Decision:**

Accept (poster)

**Comment:**

Reviewers were excited by the paper's motivation of developing methodology for structure learning in cyclic systems, in this case gene-regulatory networks (GNRs). They found the proposed methodology, based on GFlowNets, to be natural and innovative and were impressed by the method's empirical performance. While they were hesitant about some of the method's assumptions (most notably, the causal sufficiency assumption) as they pertained to GNRs, the reviewers nevertheless felt like ideas in the paper would have significant impact in the GNR literature. All of the reviewers commented on the paper's readability, which detracted from both their understanding and ultimate evaluation. While the reviewers were able to understand enough to advocate for the paper, I encourage the authors to revise the camera-ready for an audience that is less familiar with GFlowNets.